# Trends in the socioeconomic patterning of overweight/obesity in India: a repeated cross-sectional study using nationally representative data

Shammi Luhar,[1] Poppy Alice Carson Mallinson,[2] Lynda Clarke,[1] Sanjay Kinra[2]

[1]Department of Population Health, London School of Hygiene & Tropical Medicine, London, UK
[2]Department of Non-Communicable Disease Epidemiology, London School of Hygiene & Tropical Medicine, London, UK

**Correspondence to**
Shammi Luhar;
shammi.luhar@lshtm.ac.uk

## ABSTRACT

**Objectives** We aimed to examine trends in prevalence of overweight/obesity among adults in India by socioeconomic position (SEP) between 1998 and 2016.

**Design** Repeated cross-sectional study using nationally representative data from India collected in 1998/1999, 2005/2006 and 2015/2016. Multilevel regressions were used to assess trends in prevalence of overweight/obesity by SEP.

**Setting** 26, 29 and 36 Indian states or union territories, in 1998/99, 2005/2006 and 2015/2016, respectively.

**Participants** 628 795 ever-married women aged 15–49 years and 93 618 men aged 15–54 years.

**Primary outcome measure** Overweight/obesity defined by body mass index >24.99 kg/m$^2$.

**Results** Between 1998 and 2016, overweight/obesity prevalence increased among men and women in both urban and rural areas. In all periods, overweight/obesity prevalence was consistently highest among higher SEP individuals. In urban areas, overweight/obesity prevalence increased considerably over the study period among lower SEP adults. For instance, between 1998 and 2016, overweight/obesity prevalence increased from approximately 15%–32% among urban women with no education. Whereas the prevalence among urban men with higher education increased from 26% to 34% between 2005 and 2016, we did not observe any notable changes among high SEP urban women between 1998 and 2016. In rural areas, more similar increases in overweight/obesity prevalence were found among all individuals across the study period, irrespective of SEP. Among rural women with higher education, overweight/obesity increased from 16% to 25% between 1998 and 2016, while the prevalence among rural women with no education increased from 4% to 14%.

**Conclusions** We identified some convergence of overweight/obesity prevalence across SEP in urban areas among both men and women, with fewer signs of convergence across SEP groups in rural areas. Efforts are therefore needed to slow the increasing trend of overweight/obesity among all Indians, as we found evidence suggesting it may no longer be considered a 'diseases of affluence'.

## INTRODUCTION

Overweight and obesity present considerable challenges to the maintenance of global health improvements due to its association

### Strengths and limitations of this study

► Our use of the most recent nationally representative data available for Indian adults make our results the most up-to-date estimates of the socioeconomic patterning of overweight/obesity, and their trends, in India.

► Using a large nationally representative data set also enabled us to generate both precise and nationally generalisable overweight/obesity prevalence trends.

► Body mass index was the only measure used to define overweight/obesity, and prevalence estimates may vary based on the adiposity measure used and the cut-offs used. However, we would not expect the reported socioeconomic patterning of overweight/obesity, and trends, to change considerably between measures.

► Our results may mask subnational variation in overweight/obesity prevalence and trends, especially given large subnational differences in economic growth, demography and culture between India's states.

with many non-communicable diseases (NCDs).[1] WHO's aim to reduce global obesity to 2010 levels by 2025[1] is threatened by the increasing prevalence of overweight and obesity in India,[2] where nearly a sixth of the global population lives.[3]

In India, economic growth and rising incomes have been accompanied by increases in the proportion of Indians classified as overweight or obese. The proportion of adult women classified as either overweight or more than doubled for adult women from 9% to 21% between 1998 and 2016, while increasing from 11% to 19% among adult men between 2005 and 2016.[2 4 5] At the same time, undernutrition and infectious diseases continue to threaten population health,[6–9] presenting dilemmas about the appropriate allocation of scarce public finances and policy attention.

In low-income countries, overweight and obesity is usually more prevalent among higher socioeconomic position (SEP) groups,[2 10–13] whereas the opposite is observed in most high-income countries, where lower SEP individuals are more likely to be overweight or obese.[10 13] Although considered a lower middle-income country,[14] India has experienced considerable economic growth between 1998 and 2015,[15] and how this has impacted the proportion classified as overweight or obese in different SEP groups is unknown.

In this study, we aim to estimate recent trends in the proportion of Indians considered overweight or obese by SEP in India. Our results are intended to inform health policy decisions by identifying groups currently most at risk of being overweight or obese and those who have experienced the largest increases in prevalence between 1998 and 2016.[16] We hypothesise that between 1998 and 2016, the proportion classified as overweight or obese has increased in all SEP groups, in both urban and rural areas, however, with greater increases among lower SEP individuals than higher SEP individuals.

## METHODS
### Study population
The National Family Health Surveys (NFHS) 2, 3 and 4, collected in 1998–1999, 2005–2006 and 2015–2016, respectively, gathered health and demographic data on 89 199, 124 385 and 699 686 eligible women in surveys 2, 3 and 4, respectively, in addition to 74 369 and 112 122 eligible men in surveys 3 and 4, respectively.[2 4 5] As NFHS-2 only collected data on ever-married women, we restricted the sample across surveys to this population to allow comparability over time. Pregnant women were not included in our analysis as their pregnancy may bias their assessment of weight status. From this restricted sample, we further excluded women (1998–1999: n=6182 (7.4%); 2005–2006: n=3673 (4.2%); 2015–2016: 7810 (1.6%)) and men (2005–2006: n=5160 (6.8%); 2015–2016: n=3422 (3.1%)) with missing height and weight data. The analytic sample used in our main analysis consisted of 628 795 women aged 15–49 years and 93 618 men aged 15–54 years across all three surveys, representing respondents with complete data across all the key variables. In each of the surveys, multistage sampling approaches were adopted, and sampling weights were provided in the data sets.[2 4 5] Between surveys, the number of states, or union territories, in India increased from 26 in 1998–99 in to 36, due to the creation of new states from existing ones, for instance, the creation of Jharkhand from Bihar, and Telangana from Andhra Pradesh.

### Outcome
In each survey, the participants' height and weight were measured and used to calculate body mass index (BMI). To make the interpretation of our results more straightforward, we categorised the continuous BMI variable using a meaningful qualitative cut-off that facilitate comparison

with other studies and adequately capture excess adiposity. Overweight, as well as obese, adults have been reported to be at higher risk of NCDs and all cause-mortality,[17 18] therefore we categorised individuals as either overweight/obese (BMI over $24.99 \, \text{kg/m}^2$), or not overweight/obese (BMI less than or equal to $24.99 \, \text{kg/m}^2$), based on the WHO definition.[1] We additionally used cut-off values recommended for use among Asian populations to verify the trends we initially identified,[19] whereby individuals with a BMI greater than $22.99 \, \text{kg/m}^2$ were classified as overweight/obese and included the results in the online supplementary appendix. Lower BMI cut-off values may be more appropriate among Asian populations, given a potentially higher risk of overweight/obesity related diseases at lower BMI levels compared with populations on which initial classifications were based.[19]

### Independent variables
We considered two measures of SEP: an index of standard of living (SoL) and educational attainment. It was not possible to include occupation as an independent variable because it was collected on a limited subsample of respondents in the 2015–2016 survey.

We allocated individuals in all the surveys to one of the following four education categories, based on the number of years of schooling: none (0 years), primary (1–5 years), secondary (6–12 years) and higher (12+ years). We used education as a measure of SEP as it may indicate employable skills that expose individuals to more opportunities to earn higher incomes.

The NFHS contains a wealth index, constructed using Principal Components Analysis (PCA) in each survey separately, using information on household asset ownership and household characteristics. As the original wealth index cannot be appropriately compared over time, and as we intended to stratify our analysis by urban and rural areas, we constructed a new index, as an alternative measure of SEP, using PCA from 26 assets and characteristics available in all the surveys.[2 4 5] Based on our new wealth scores derived from weightings given to each asset or characteristic, households were classified as either 'lower', 'medium' or 'higher' SoL. Asset-based indices are commonly used in cross-sectional studies conducted in low-income and middle-income countries, where income data may be an unreliable indicator of overall SEP, particularly in rural areas.[20] For instance, households may receive income from a variety of sources, which may be difficult to recall, or income may be received in kind[20 21] rather than monetarily. Consequently, a household's stock of assets may provide a more reliable measure of current SEP.[20]

We adjusted our final models for the respondent's age (categorised as 15–29 years, 30–39 years and 40–49 years (40–54 years) for women (men)), as it has been reported in previous studies that overweight/obesity prevalence increases with age.[22] Additionally, older adults may have accumulated more assets over a longer lifespan, potentially, confounding the association between SEP and

overweight/obesity. Research has found overweight/obesity to be higher among married individuals, and therefore could confound the reported association between SEP and overweight/obesity.

## Statistical analysis

We initially calculated the prevalence of overweight/obesity in each SoL index and educational attainment category by sex and urban/rural residence. We accounted for the complex survey design of the data using sampling weights. Separately for urban and rural areas, we calculated the ratio of the prevalence between the highest and lowest socioeconomic status group of our two main SEP variables (eg, higher to lower SoL, and higher to no education) in each of the surveys. Additionally, we calculated the percentage change in the prevalence of overweight/obesity by each category of SoL and educational attainment.

Separately for urban and rural areas and sex, we fitted multilevel logistic regression models with random intercepts for primary sampling units and states. We chose to include Primary Sampling Unit (PSU)-level and state-level random intercepts due to the hierarchical nature of the NFHS data, whereby individuals are nested within PSUs, which are nested within states. SEs calculated in our models would have been underestimated if we did not account for this clustering. We modelled the log OR of overweight/obesity in each category of the SEP variable of interest in each of the surveys by fitting a survey-specific interaction term. The regression models were adjusted for the covariates mentioned in the independent variables section, in addition to the remaining SEP variable. No evidence of multicollinearity of independent variables with the main exposure of interest was detected when examining changes in the SE once new variables were added. Finally, we derived and reported the predicted prevalence of overweight/obesity from the model, in addition to their 95% confidence bounds. Adjusted analyses were also carried out using Asian specific BMI cut-offs to observe if the trends identified varied depending on the outcome measure used (online supplementary appendix).

## Patient and public involvement

Publicly available survey data were used for the analysis, and no patients were involved in the study.

## RESULTS

The study population generally experienced increasing educational attainment and SoL over the period of analysis in both urban and rural areas. Whereas the percentage of respondents with no education declined over the study period, particularly among the rural population, the percentage with secondary education in the 2015–2016 survey was generally higher than in 1998–1999 and 2005–2006. Additionally, in both rural and urban areas, the percentage of individuals from lower SoL households declined, while the percentage from higher SoL households increased between 1998 and 2016 (tables 1 and 2).

The prevalence of overweight/obesity increased in each successive survey for both of our samples of men and women. In rural India, the prevalence among men almost tripled from 0.059 to 0.148 between 2005 and 2016, and

**Table 1** Characteristics of rural study participants across NFHS surveys with recorded BMI information

| | Women | | | | | | Men | | | |
| | NFHS 2 (1998–1999) | | NFHS 3 (2005–2006) | | NFHS 4 (2015–2016) | | NFHS 3 (2005–2006) | | NFHS 4 (2015–2016) | |
| | Freq | Proportion | Freq | Proportion | Freq | Proportion | Freq | Proportion | Freq | Proportion |
|---|---|---|---|---|---|---|---|---|---|---|
| Not overweight/obese | 49 596 | 0.93 | 42 979 | 0.9 | 2 89 482 | 0.83 | 32 304 | 0.93 | 64 133 | 0.86 |
| Overweight/obese | 3496 | 0.07 | 4912 | 0.1 | 61 124 | 0.17 | 2255 | 0.07 | 10 550 | 0.14 |
| Age 15–29 years | 23 888 | 0.45 | 19 279 | 0.4 | 1 26 796 | 0.36 | 16 537 | 0.48 | 34 589 | 0.46 |
| Age 30–39 years | 17 488 | 0.33 | 16 892 | 0.35 | 1 22 520 | 0.35 | 8951 | 0.26 | 18 965 | 0.25 |
| Age 40-49 years (54 males) | 11 716 | 0.22 | 11 720 | 0.24 | 1 01 290 | 0.29 | 9071 | 0.26 | 21 129 | 0.28 |
| No education | 31 724 | 0.6 | 24 314 | 0.51 | 1 46 302 | 0.42 | 6904 | 0.2 | 11 709 | 0.16 |
| Primary | 9469 | 0.18 | 8417 | 0.18 | 55 652 | 0.16 | 6620 | 0.19 | 10 545 | 0.14 |
| Secondary | 9971 | 0.19 | 13 872 | 0.29 | 1 31 722 | 0.38 | 18 199 | 0.53 | 43 737 | 0.59 |
| Higher | 1916 | 0.04 | 1285 | 0.03 | 16 930 | 0.05 | 2824 | 0.08 | 8692 | 0.12 |
| Low SoL | 28 408 | 0.54 | 21 262 | 0.44 | 64 998 | 0.19 | 14 615 | 0.42 | 11 842 | 0.17 |
| Middle SoL | 18 616 | 0.35 | 15 929 | 0.33 | 1 20 050 | 0.36 | 12 508 | 0.36 | 25 338 | 0.35 |
| High SoL | 5869 | 0.11 | 10 645 | 0.22 | 1 49 191 | 0.45 | 7409 | 0.21 | 34 202 | 0.48 |
| Married | 49 674 | 0.94 | 44 763 | 0.93 | 3 31 883 | 0.95 | 22 352 | 0.65 | 47 948 | 0.64 |
| Not married | 3418 | 0.06 | 3128 | 0.07 | 18 723 | 0.05 | 12 207 | 0.35 | 26 735 | 0.36 |

BMI, body mass index; NFHS, National Family Health Surveys; SoL, standard of living.

**Table 2**  Characteristics of urban study participants across NFHS surveys with recorded BMI information

|  | NFHS 2 (1998–1999) | | NFHS 3 (2005–2006) | | NFHS 4 (2015–2016) | | NFHS 3 (2005–2006) | | NFHS 4 (2015–2016) | |
|---|---|---|---|---|---|---|---|---|---|---|
|  | Freq | Proportion | Freq | Proportion | Freq | Proportion | Freq | Proportion | Freq | Proportion |
| Not overweight/ obese | 18 473 | 0.75 | 25 454 | 0.7 | 87 695 | 0.64 | 28 669 | 0.83 | 25 285 | 0.74 |
| Overweight/obese | 6048 | 0.25 | 10 808 | 0.3 | 49 443 | 0.36 | 5981 | 0.17 | 8732 | 0.26 |
| Age 15–29 years | 8950 | 0.36 | 12 401 | 0.34 | 41 893 | 0.31 | 17 434 | 0.5 | 15 581 | 0.46 |
| Age 30–39 years | 9253 | 0.38 | 13 954 | 0.38 | 52 032 | 0.38 | 8652 | 0.25 | 8742 | 0.26 |
| Age 40–49 (54 males) | 6318 | 0.26 | 9907 | 0.27 | 43 213 | 0.32 | 8564 | 0.25 | 9694 | 0.28 |
| No education | 6493 | 0.26 | 9048 | 0.25 | 28 878 | 0.21 | 3016 | 0.09 | 2884 | 0.08 |
| Primary | 4025 | 0.16 | 4959 | 0.14 | 16 818 | 0.12 | 4143 | 0.12 | 3407 | 0.1 |
| Secondary | 8814 | 0.36 | 16 655 | 0.46 | 67 583 | 0.49 | 19 902 | 0.57 | 19 563 | 0.58 |
| Higher | 5181 | 0.21 | 5596 | 0.15 | 23 859 | 0.17 | 7574 | 0.22 | 8163 | 0.24 |
| Low SoL | 16 444 | 0.67 | 17 263 | 0.48 | 33 609 | 0.25 | 17 329 | 0.5 | 8773 | 0.27 |
| Middle SoL | 5682 | 0.23 | 10 147 | 0.28 | 50 027 | 0.38 | 9613 | 0.28 | 11 925 | 0.36 |
| High SoL | 2310 | 0.09 | 8832 | 0.24 | 49 540 | 0.37 | 7694 | 0.22 | 12 389 | 0.37 |
| Married | 22 931 | 0.94 | 33 845 | 0.93 | 128 279 | 0.94 | 19 656 | 0.57 | 20 375 | 0.6 |
| Not married | 1590 | 0.06 | 2417 | 0.07 | 8859 | 0.06 | 14 994 | 0.43 | 13 642 | 0.4 |

BMI, body mass index; NFHS, National Family Health Surveys; SoL, standard of living.

among women, the prevalence increased from 0.059 to 0.182 between 1998 and 2016. In urban India, the prevalence among women increased to 0.385 in 2015–2016, from 0.236 in 1998–1999, whereas the prevalence among urban men increased from 0.167 to 0.276 between 2005 and 2016 (figure 1).

In all surveys, and for men and women in both urban and rural areas, the prevalence of overweight/obesity was

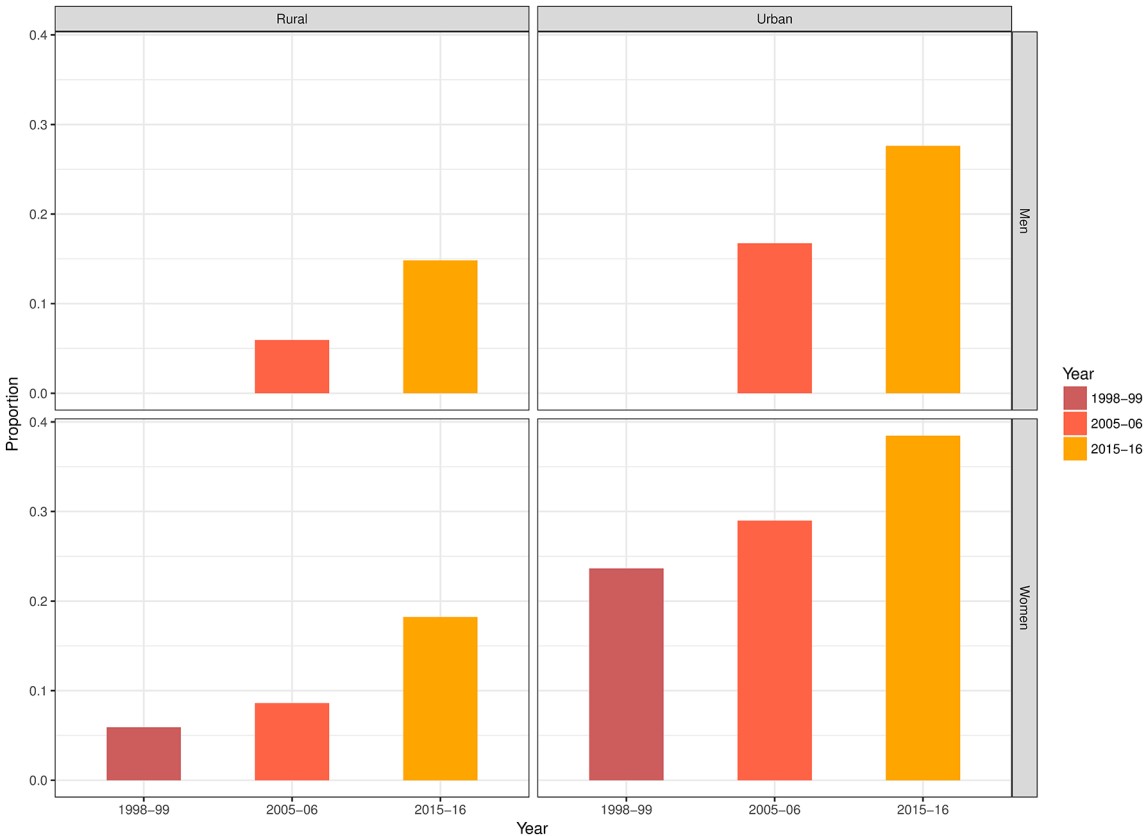

**Figure 1**  Prevalence (weighted) of overweight/obesity in urban and rural India among men and women.

**Table 3** Percentage of respondents classified as overweight/obese by education level (1998–2016)

| | Women | | | | Men | | |
|---|---|---|---|---|---|---|---|
| | 1998–1999 | 2005–2006 | 2015–2016 | % change | 2005–2006 | 2015–2016 | % change |
| | % | % | % | 1998–2016 | % | % | 2005–2016 |
| **Rural** | | | | | | | |
| Education* | | | | | | | |
| No education | 3.38 | 5.26 | 13.91 | 311.54 | 3.05 | 10.79 | 253.77 |
| Primary | 7.93 | 10.01 | 18.45 | 132.66 | 4.22 | 14.06 | 233.18 |
| Secondary | 10.8 | 14.19 | 21.82 | 102.04 | 6.57 | 14.56 | 121.61 |
| Higher | 15.85 | 22.79 | 26.73 | 68.64 | 15.32 | 22.32 | 45.69 |
| Ratio† | 4.69 | 4.33 | 1.92 | | 5.02 | 2.07 | |
| **Urban** | | | | | | | |
| Education* | | | | | | | |
| No education | 13.53 | 18.49 | 32.17 | 137.77 | 7.73 | 18.28 | 136.48 |
| Primary | 19.45 | 24.45 | 37.21 | 91.31 | 10.9 | 23.86 | 118.90 |
| Secondary | 27.18 | 33.04 | 40.15 | 47.72 | 15.24 | 26.33 | 72.77 |
| Higher | 35.35 | 41.79 | 41.56 | 17.57 | 28.39 | 34.87 | 22.82 |
| Ratio† | 2.61 | 2.26 | 1.29 | | 3.67 | 1.91 | |

*$\chi^2$ test p value of each Strata's association with overweight/obesity: p<0.001.
†Ratio of the percentage among individuals with higher education and no education.

highest among participants with higher education and from a higher SoL, whereas the lowest prevalence of overweight/obesity was found among participants with no education and from a lower SoL.

However, over the study periods for both men and women, the greatest percentage increase in overweight/obesity prevalence was observed among participants from the lowest SoL category and participants with no education. Consequently, the ratio of the prevalence of

overweight/obesity in all of the highest, compared with the lowest, SEP groups, reduced over time (tables 3 and 4).

After adjusting for marital status and age, in urban areas, the predicted prevalence of overweight/obesity among lower SEP women increased over the study period for both men and women, whereas no notable changes were observed among higher SEP women. Among urban men, we observed some increase in the prevalence

**Table 4** Percentage of respondents classified as overweight/obese by standard of living (SoL) (1998–2016)

| | Women | | | | Men | | |
|---|---|---|---|---|---|---|---|
| | 1998–1999 | 2005–2006 | 2015–2016 | % change | 2005–2006 | 2015–2016 | % change |
| | % | % | % | 1998–2016 | % | % | 2005–2016 |
| **Rural** | | | | | | | |
| SoL* | | | | | | | |
| Lower SoL | 2.35 | 3.01 | 6.65 | 182.98 | 1.79 | 4.96 | 177.09 |
| Middle SoL | 8.22 | 8.88 | 12.94 | 57.42 | 5.66 | 9.47 | 67.31 |
| Higher SoL | 22.93 | 25.15 | 27.74 | 20.98 | 17.49 | 22.3 | 27.50 |
| Ratio† | 9.76 | 8.36 | 4.17 | | 9.77 | 4.50 | |
| **Urban** | | | | | | | |
| SoL* | | | | | | | |
| Lower SoL | 16.32 | 17.36 | 24.91 | 52.63 | 8.92 | 16.01 | 79.48 |
| Middle SoL | 39.11 | 35.01 | 38.83 | –0.72 | 20.61 | 26.89 | 30.47 |
| Higher SoL | 46.93 | 48.4 | 46.87 | –0.13 | 30.59 | 35.77 | 16.93 |
| Ratio† | 2.88 | 2.79 | 1.88 | | 3.43 | 2.23 | |

*$\chi^2$ test p value of each strata's association with overweight/obesity: p<0.001.
†Ratio of the percentage in the highest and lowest socioeconomic group.

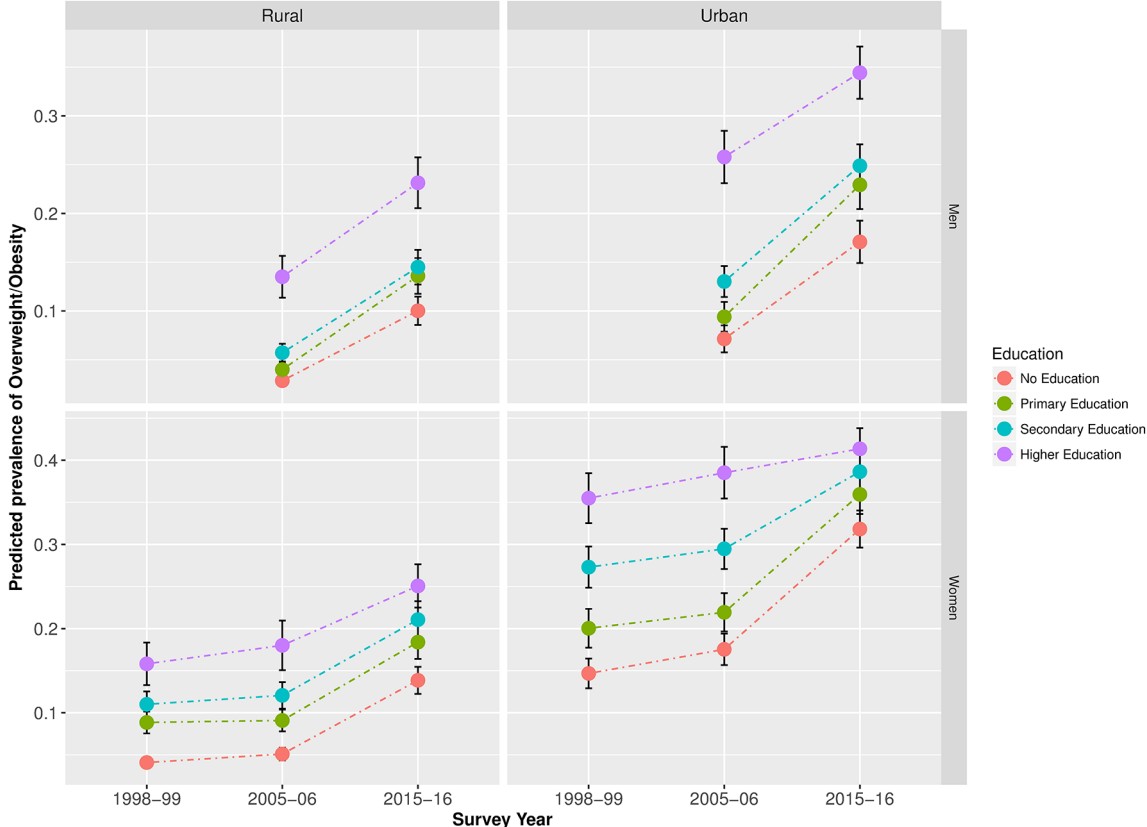

**Figure 2** Predicted prevalence* of overweight/obesity in India by educational attainment (1998–2016).

of overweight/obesity among high SEP respondents; however, the increase among low SEP men was greater. Among both rural men and women, more similar increases were observed among individuals from all SEP groups over the study period (figures 2 and 3). Equivalent trends were found when using the BMI cut-offs recommended for Asian populations (online figures A1 and A2 in supplementary appendix).

## DISCUSSION

We found that, although overweight/obesity prevalence increased with SEP, in urban areas no notable change in the prevalence of overweight/obesity was observed among higher SEP women, whereas the prevalence among lower SEP women increased considerably between 1998 and 2016. The prevalence increase of overweight/obesity was greater among lower SEP urban men compared with higher SEP counterparts between 2005 and 2016. Consequently, some convergence of overweight/obesity across SEP was observed in urban areas among both men and women. In rural areas, however, overweight/obesity prevalence increased similarly among individuals in all SEP groups, with fewer signs of convergence across SEP groups yet.

### Strengths and limitations

The main strength of our study is our use of the most recent nationally representative data available for India,

making our results the most up-to-date estimates of overweight/obesity trends by SEP.

Our study however has some limitations. First, we derive our only measure of overweight/obesity from BMI, rather than complement our results with alternative measures of overweight/obesity, such as waist circumference[23 24] and body fat percentage. Consequently, prevalence estimates may vary depending on the adiposity measure and the exact definitions/cut-offs used. However, given the high correlation between BMI and measures including waist circumference among Indians,[25] we would not expect the reported associations between overweight/ obesity and SEP, and trends, to change considerably between measures.

Second, to ensure the population of sampled women was comparable over time, we limited our analysis to ever-married women, as this was the selection criteria in the NFHS-2 survey. Prevalence of overweight/obesity is generally lower among never-married women,[26] for instance in the NFHS-4 survey data, the prevalence of overweight/obesity was 6.6% among never-married women, compared with 25.0% among currently married women. This may have lead us to overestimate overweight/obesity prevalence among women, as the weighted percentage of never-married women were 19.8% and 22.5% in the 2005–2006 and 2015–2016 samples, respectively. However, although individual point estimates may be affected, we do not expect the

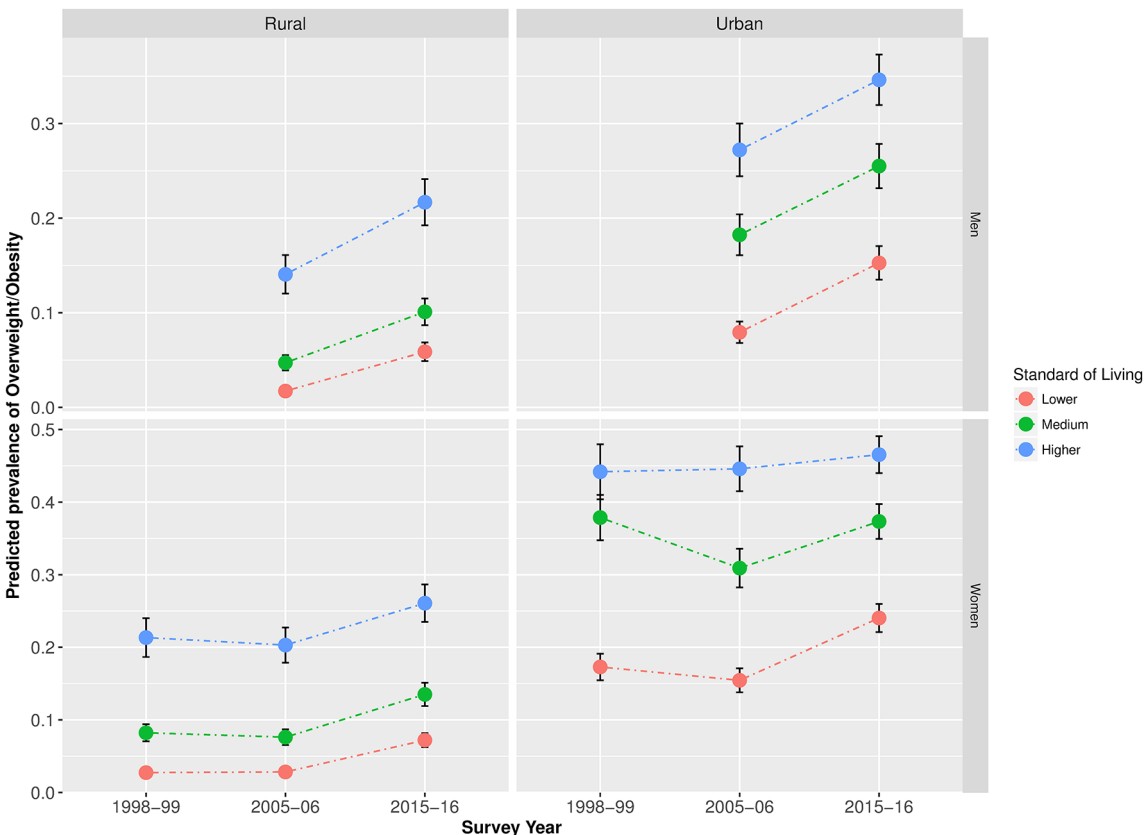

**Figure 3** Predicted prevalence* of overweight/obesity in India by standard of living (1998–2016).

association between overweight/obesity and SEP we identified to be overestimated.

Our SoL index may also imperfectly capture household wealth. For instance, no indication about the quality of assets used in the measure were included, potentially misclassifying certain households.[20 27] However, as three broad SoL groups across a large data set were defined, we do not expect any misclassification to substantially bias our results. Additionally, the association between the true SEP and certain assets included in the SoL index may differ between urban and rural areas. We attempted to account for differences in the value of certain assets by calculating separate indices for urban and rural areas; however, differences in the value of some assets may still exist within broader geographical areas, for instance between states.

Finally, our results may mask variation in subnational prevalence and trends, especially given subnational differences between states in economic growth, demography and culture. For instance, research in India has found that in states with a higher prevalence of overweight, lower and higher SEP group may show a converging risk of overweight/obesity, whereas divergent trends have been identified in states with the highest proportion of underweight individuals.[28]

### Comparison with other research

The only other India-specific national study we found on this topic did not identify any change in the overweight/

obesity-SEP association between 1998–1999 and 2005–2006 in urban or rural India, with a persisting higher prevalence among high SEP groups.[29] Beyond 2005–2006, the authors predicted that future overweight/obesity prevalence would show a similar social patterning as they expected future economic gains to almost solely benefit higher SEP individuals. By contrast, the converging socioeconomic patterning of overweight/obesity we have identified in urban areas indicates that economic growth in the past decade may either have been more egalitarian than previously expected, the cost of high calorie food may have become less expensive or even the pool of susceptible higher SEP individuals may be becoming saturated.

Converging overweight/obesity prevalence between higher and lower SEP groups has been identified subnationally in India, when restricted to states defined by a high overall prevalence of overweight,[28] mirroring our finding in urban areas. This may suggest that convergence is restricted to areas that have moved beyond the earliest stages of the epidemiological transition.

Though not reported in previous nationally representative studies in India, a converging socioeconomic patterning of overweight/obesity has been noted in some other low-income and middle-income countries, where the highest increases in overweight prevalence have been found among women working in manual labour,[30] among the lowest wealth and income groups[31–33] and among rural residents.[34]

## Potential mechanisms

In rural areas, we identified similar increases in prevalence among individuals from all SEP groups. Some studies suggest that in low-income settings, increases in overweight and obesity are restricted to higher SEP individuals, which may be due to changing dietary patterns towards fatty and sugary convenience foods[9–13 35]; however, the rising prevalence among lower SEP individuals indicates that they may also be increasingly exposed to high-calorie foods. Some researchers have also suggested that this mechanism is stronger in low-income or rural settings due to more favourable perceptions of large body sizes across socioeconomic status.[13 36–38]

In urban India, the greater increase in overweight/obesity prevalence among lower SEP individuals mirrors similar findings from places at relatively later stages of economic development, where some researchers have suggested that lower SEP individuals may be priced out of affording relatively expensive low-calorie healthy diets.[13 39–41] Additionally, lower SEP individuals in urban areas may be more exposed to sedentary lifestyles driven by technological advances replacing manual energy-exerting labour and improved transport links.[42 43] Increased health consciousness, in combination with the ability to afford low calorie diets, may explain why no notable change in overweight/obesity prevalence among the higher SEP urban population was found[13 44 45] in addition to the potential saturation of individuals susceptible to becoming overweight or obese.

## Implications

Some studies argue that in India, NCD risk factors are almost exclusively an issue for higher SEP individuals.[46] However, our finding that overweight/obesity prevalence has increased among lower SEP individuals in both urban and rural areas implies that to consider overweight/obesity as 'diseases of affluence'[47] may not be appropriate in India's current context. Efforts to tackle the overall increasing overweight/obesity trend must be inclusive of both the urban and rural poor. This may be especially urgent due to the compounding effect of overweight/obesity and associated NCDs on infectious diseases, which are still highly prevalent among the poor.

Recent initiatives to raise population health include the launch of an integrated National Health Mission,[48] which aims to address deficiencies in healthcare delivery across the socioeconomic spectrum in urban and rural areas. Such initiatives may benefit from information about the increasing prevalence among low SEP Indians, as future action aimed at preventing overweight and obesity can be targeted accordingly. Due to the positive association of overweight and obesity with NCDs such as stroke and diabetes,[49 50] urgency is required in addressing this modifiable risk factor especially as it could compound existing health complications among poorer Indians, where communicable disease and undernutrition-related diseases already tend to be more prevalent.

## CONCLUSION

Although India is still considered as a lower middle-income country, we have identified some convergence of overweight/obesity prevalence across SEP in urban areas among both men and women, with fewer signs of convergence across SEP groups in rural areas. Our findings suggest that an urgent response is needed to slow the increasing trend among poorer Indians, particularly as increasing exposure to overweight and obesity related diseases may compound an already high exposure to infectious diseases.

**Acknowledgements** We would like to thank measure DHS for granting us access to the data used in this study.

**Contributors** The authors' responsibilities were as follows: SL and SK designed the study; SL performed the data analysis and takes responsibility for the final content; SL interpreted the results; SL drafted the manuscript; PACM, LC and SK reviewed and approved the final manuscript.

**Funding** This study was funded by the Economic and Social Research Council (grant number: ES/J500021/1).

**Competing interests** None declared.

**Patient consent** Not required.

**Ethics approval** The analysis of secondary data was approved by the London School of Hygiene & Tropical Medicine's Research Ethics Committee.

**Provenance and peer review** Not commissioned; externally peer reviewed.

**Data sharing statement** All datasets in this analysis are available at http://www.measuredhs.com.

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
