## [Reviewer comments · BMJ Open]

ARTICLE DETAILS

TITLE (PROVISIONAL)	Trends in the socioeconomic patterning of overweight/obesity in India: a repeated cross-sectional study using nationally representative data
AUTHORS	Luhar, Shammi; Mallinson, Poppy; Clarke, Lynda; Kinra, Sanjay

VERSION 1 – REVIEW

REVIEWER	Helen Booth Clinical Practice Research Datalink, UK
REVIEW RETURNED	24-May-2018

GENERAL COMMENTS	I found the topic of the paper interesting but have some comments/questions that I think should be addressed prior to publication. Overall, I think the manuscript would be improved by further explanation of why the authors chose to combine overweight/obesity as their sole outcome, and also to explain why the statistical method chosen was the most appropriate to answer this question. P2 Abstract: L16 The participants are given as ever-married women and men, but information later in the paper leads me to understand that this only applied to women in the study. L28-37 No results have been presented in this section of the abstract. Without any inclusion of numbers this reads this section is not convincing. The authors should include actual study findings to support their descriptions. L40-43 The study conclusions do not relate clearly to the results section. P4 Methods L21-31 It wasn't clear why you decided to focus on overweight/obesity combined rather than obesity alone or looking at BMI as a linear variable. I think it would be helpful to add further detail here. The lower cut-off for Asian populations is discussed but it wasn't been specified whether you actually used these for the sensitivity analyses. L36-37 Without reading about the surveys in depth it wasn't clear whether you chose the independent variables as the only markers of SES available or whether there were a wider number from which you chose those as the most appropriate. I would find it useful if this was reworded to make that clear. P5 L13-15 It's not very clear what the association of marital status is with obesity based on the current wording.
--

	P33-45 I'm not very familiar with the statistical method you used and I think the paper would benefit from a greater explanation as to why this was the most appropriate method to answer your research question. Was multilevel modelling necessary? I have seen other similar prevalence studies based on survey data that do not use multilevel modelling. How did you choose your levels (primary sampling units and states if I understood correctly)? Were these appropriate in size for use in the model? What were the states? – it would be helpful to have detail on these. General - I couldn't see any information on how you dealt with missing information in the study. It's unlikely that the survey had complete data for weight and height in a study of this scale. Please add further details on how this was dealt with. P6 Table 1 The rural/urban split hasn't been presented in isolation (only in conjunction with the SoL data). I think it would be helpful to see this data separately. P9 Discussion L12 I think there is a typo '...overweight/obesity is generally higher among never-married women..'. Should this read lower?
--	--

REVIEWER	Dr. Tehzeeb Zulfiqar Australian National University, Australia
REVIEW RETURNED	26-Jun-2018

GENERAL COMMENTS	It is a very important study as there are gaps in evidence from low-and-middle-income countries about obesity prevalence. My comments are as below;  1. Introduction. Para 1 line 24 & 3 line 43. The term over nutrition is confusing. Obesity would be more appropriate term to use here. 2. Please use one term for overweight/obesity; either use overweight or obese OR overweight/obese OR overweight and obese. At present all three terms are used in the introduction. 3. The term "waves" give an impression that NFHS is a longitudinal study. If so, then why longitudinal analysis was not done? If it is a repeated cross-sectional study then please use another term instead of waves. 4. The second sentence of results (For instance, whereas the percentage of women in the sample classified as having no education....is quite confusing. please try to rewrite. In addition, does " the previous ones" in the end of the sentence mean the previous surveys? 5. Please revise the sentence in 2nd para line 33 of "potential mechanisms" for clarity (..some researchers have suggested that high prices of low calorie foods may price lower SEP individuals out of healthy diets) 6. Reference 23 line 52 page 12, is in a different format than the rest. Please check.
--

VERSION 1 – AUTHOR RESPONSE

Reviewer: 1

Reviewer Name: Helen Booth

Institution and Country: Clinical Practice Research Datalink, UK

Please state any competing interests or state 'None declared': None declared

I found the topic of the paper interesting but have some comments/questions that I think should be addressed prior to publication. Overall, I think the manuscript would be improved by further explanation of why the authors chose to combine overweight/obesity as their sole outcome, and also to explain why the statistical method chosen was the most appropriate to answer this question.

Thank you for your helpful comments and suggestions.

P2 Abstract:

L16 The participants are given as ever-married women and men, but information later in the paper leads me to understand that this only applied to women in the study.

Response: Apologies, this was a typo in the abstract and has been amended accordingly.

“628,795 ever-married women aged 15–49 years and 93,618 men aged 15-54.”

L28-37 No results have been presented in this section of the abstract. Without any inclusion of numbers this reads this section is not convincing. The authors should include actual study findings to support their descriptions.

Response: We agree that the abstract does benefit from the inclusion of numbers relating to study findings. We have amended the results section of the abstract accordingly:

“Between 1998 and 2016, overweight/obesity prevalence increased among men and women in both urban and rural areas. In all periods, overweight/obesity prevalence was consistently highest among higher SEP individuals. In urban areas, overweight/obesity prevalence increased considerably over the study period among lower SEP adults. For instance, between 1998 and 2016, overweight/obesity prevalence increased from approximately 15% to 32% among urban women with no education. Whereas the prevalence among urban men with higher education increased from 26% to 34% between 2005 and 2016, we did not observe any notable changes among high SEP urban women between 1998 and 2016. In rural areas, more similar increases in overweight/obesity prevalence were found among all individuals across the study period, irrespective of SEP. Among rural women with higher education, overweight/obesity increased from 16 to 25% between 1998 and 2016, whilst the prevalence among rural women with no education increased from 4% to 14%.”

L40-43 The study conclusions do not relate clearly to the results section.

Response: Agreed. Our initial take home conclusion was that the growing prevalence among lower SEP Indians was considerable and counter to what may be expected in a lower middle-income country like India. However, we have amended the section to allude to the fact that overweight/obesity does not solely affect the affluent in India's current context.

“Efforts are therefore needed to slow the increasing trend of overweight/obesity among all Indians, as we found evidence suggesting it may no longer be considered a ‘diseases of affluence’.”

P4 Methods

L21-31 It wasn't clear why you decided to focus on overweight/obesity combined rather than obesity alone or looking at BMI as a linear variable. I think it would be helpful to add further detail here. The lower cut-off for Asian populations is discussed but it wasn't been specified whether you actually used these for the sensitivity analyses.

Response: Thank you for this comment; we agree that this section would benefit from further clarification. One reason we chose not use BMI in its continuous form was that the central theme of this paper was to examine the socioeconomic patterning of excess adiposity, which are likely to have negative public health consequences, given its association with NCDs. In a low middle-income country like India, where a considerable proportion of the population is undernourished, increasing trends in BMI we would expect to observe could be driven by a combination of people transitioning into overweight BMI groups, as well as people transitioning from an underweight to normal weight BMI classification (the latter not being something we intended to capture in this study). We believe categorising BMI according to a meaningful cut-off better captured the adverse effects of increasing overweight/obesity. Additionally, using cut-offs enable comparison with studies that have attempted to answer similar questions and make our results much more directly interpretable than the BMI value.

Our primary reason for using a binary overweight/obesity outcome was to enhance the clarity of our central message. With two categorical exposure variables, and stratification of analysis by sex and rural/urban residence, analysing overweight and obesity separately would have led to a large number of graphs, which we felt would obscure the main message of the paper. Based on our binary classification there is a clear, policy-relevant message about recent increases in excess adiposity in India and its evolving SEP patterning, and adding extra BMI categories would not greatly strengthen this message. Other studies have adopted similar approaches (Siddiqui and Donato 2015; Tuoyire et al 2016). Finally, as mentioned in the manuscript, all cause-mortality has been found to increase among people with BMI \geq 25 (Aune et al 2016; Di Angelantonio et al 2016), making this the key cut-off of public health concern marking excess adiposity.

References

Siddiqui, M.Z. and Donato, R., 2015. Overweight and obesity in India: policy issues from an exploratory multi-level analysis. *Health policy and planning*, 31(5), pp.582-591.

Tuoyire, D.A., Kumi-Kyereme, A. and Doku, D.T., 2016. Socio-demographic trends in overweight and obesity among parous and nulliparous women in Ghana. *BMC obesity*, 3(1), p.44.

Aune, D., Sen, A., Prasad, M., Norat, T., Janszky, I., Tonstad, S., Romundstad, P. and Vatten, L.J., 2016. BMI and all cause mortality: systematic review and non-linear dose-response meta-analysis of 230 cohort studies with 3.74 million deaths among 30.3 million participants. *bmj*, 353, p.i2156.

Di Angelantonio, E., Bhupathiraju, S.N., Wormser, D., Gao, P., Kaptoge, S., de Gonzalez, A.B., Cairns, B.J., Huxley, R., Jackson, C.L., Joshy, G. and Lewington, S., 2016. Body-mass index and all-cause mortality: individual-participant-data meta-analysis of 239 prospective studies in four continents. *The Lancet*, 388(10046), pp.776-786.

“In each survey, the participants’ height and weight were measured and used to calculate Body Mass Index (BMI). **To make the interpretation of our results more straightforward, we categorised the continuous BMI variable using a meaningful qualitative cut-off that facilitate comparison with other studies and adequately capture excess adiposity. Overweight, as well as obese, adults have been reported to be at higher risk of NCDs and all cause-mortality, therefore, we** categorised individuals as either overweight/obese (BMI over 24.99kg/m²), or not overweight/obese (BMI less than or equal to 24.99kg/m²), based on the WHO definition.”

We supplemented our main findings by also using the Asian BMI cut-offs in final regressions, and included the results in the Appendix. We amended the ‘Outcome’ section of the manuscript to explicitly refer to where the results of that analysis can be found.

“... whereby individuals with a BMI greater than 22.99kg/m² were classified as overweight/obese, **and included the results in the Appendix A.**”

L36-37 Without reading about the surveys in depth it wasn’t clear whether you chose the independent variables as the only markers of SES available or whether there were a wider number from which you chose those as the most appropriate. I would find it useful if this was reworded to make that clear.

Other measures of SEP are included in the data set, in particular ‘respondent’s occupation’, however the most widely used ones, especially when analysing DHS data, are a wealth index built using PCA and the respondent’s education. We initially did consider ‘occupation’ but decided against it due to data sparsity issues in the NFHS 4 survey. Additionally, it has been suggested that occupation is less informative than other indicators in settings such as India where women have low labour force participation rates (<30% in 2015) (Galobardes et al 2006; World Bank, 2015).

We outline the reasons as to why we chose to use SoL index and Education in their respective paragraphs, and rephrased the opening paragraph of the 'Independent variables' section to highlight that the respondent's occupation was initially considered.

References

Galobardes, B., Shaw, M., Lawlor, D.A., Lynch, J.W. and Smith, G.D., 2006. Indicators of socioeconomic position (part 1). *Journal of Epidemiology & Community Health*, 60(1), pp.7-12.

World Bank, 2015. Labor force participation rate, female (% of female population ages 15+)(modeled ILO estimate).

“We considered two measures of SEP: an index of standard of living and educational attainment. It was not possible to include occupation as an independent variable because it was collected on a limited subsample of respondents in the 2015-16 survey.”

P5

L13-15 It's not very clear what the association of marital status is with obesity based on the current wording.

Response: Agreed. The initial wording did not indicate the directionality of the association in the literature between marriage and obesity. We changed the sentence to the following:

“Research has found overweight/obesity to be higher among married individuals and therefore could confound the reported association between SEP and overweight/obesity.”

P33-45 I'm not very familiar with the statistical method you used, and I think the paper would benefit from a greater explanation as to why this was the most appropriate method to answer your research question. Was multilevel modelling necessary? I have seen other similar prevalence studies based on survey data that do not use multilevel modelling. How did you choose your levels (primary sampling units and states if I understood correctly)? Were these appropriate in size for use in the model? What were the states? – it would be helpful to have detail on these.

Response: The NFHS was collected using a multistage sampling approach. Multilevel modelling is used when there is non-independence of observations within clusters, which violates a key assumption of standard statistical models. Failing to account for clustering of the data can underestimate standard errors and therefore confidence intervals. We used PSU as a level as it was used as a cluster in the multistage sampling strategy and we found evidence of a substantial clustering effect at this level. We additionally included states as a higher level random effect because

we expected substantial variation between the states (which are politically, culturally and geographically distinct). There around 30 states in India (varying between the survey years, as explained below), generally considered the minimum number of clusters appropriate for multilevel models (Kreft 1996). Multilevel models accounting for clustering at PSU and state levels have been adopted in a number of papers using the NFHS in India (eg. Subramanian et al 2007; Subramanian et al 2006; Ackerson et al 2008; Balarajan et al 2013).”

References

Kreft, I.G., 1996. Are multilevel techniques necessary? An overview, including simulation studies. *Unpublished manuscript, California State University, Los Angeles.*

Subramanian, S.V., Kawachi, I. and Smith, G.D., 2007. Income inequality and the double burden of under-and overnutrition in India. *Journal of Epidemiology & Community Health*, 61(9), pp.802-809.

Subramanian, S.V., Nandy, S., Irving, M., Gordon, D., Lambert, H. and Davey Smith, G., 2006. The mortality divide in India: the differential contributions of gender, caste, and standard of living across the life course. *American Journal of Public Health*, 96(5), pp.818-825.

Ackerson, L.K., Kawachi, I., Barbeau, E.M. and Subramanian, S.V., 2008. Geography of underweight and overweight among women in India: a multilevel analysis of 3204 neighborhoods in 26 states. *Economics & Human Biology*, 6(2), pp.264-280.

Balarajan, Y.S., Fawzi, W.W. and Subramanian, S.V., 2013. Changing patterns of social inequalities in anaemia among women in India: cross-sectional study using nationally representative data. *BMJ open*, 3(3), p.e002233.

The following sentences were added to address the reasons as to why the multilevel model was adopted:

“We chose to include PSU- and state-level random intercepts due to the hierarchical nature of the NFHS data, whereby individuals are nested within PSUs, which are nested within states. Standard errors calculated in our models would have been underestimated if we did not account for this clustering.”

In the ‘Study Population’ section, we also included a small section addressing states, and why it increases over the surveys in response to the point about which states were included.

“Between surveys, the number of states or union territories we included in the analysis increased from 26 in 1998-99 in to 36, due to the creation of new states from existing ones, for instance, the creation of Jharkhand from Bihar, and Telangana from Andhra Pradesh.”

General - I couldn't see any information on how you dealt with missing information in the study. It's unlikely that the survey had complete data for weight and height in a study of this scale. Please add further details on how this was dealt with.”

Response: Thank you for this comment, and apologies for its initial omission. We have included a part, with numbers, in the section headed 'Study Population', on how we dealt with missing data. In brief, respondents without height and weight data, were excluded from the sample and final regression models from which predicted prevalence were derived included respondents with full data across all the variables. As this represented a relatively small proportion of the initial sample, we did not believe it to be of particular concern, and other papers using NFHS data have adopted the same approach (Subramanian et al 2006; Subramanian et al 2009). In the manuscript the amended section is as follows:

The National Family Health Surveys (NFHS) 2, 3 and 4, collected in 1998-99, 2005-06 and 2015-16, respectively, gathered health and demographic data on 89,199, 124,385 and 699,686 eligible women in surveys 2, 3 and 4, respectively, in addition to 74,369 and 112,122 eligible men in surveys 3 and 4, respectively. **Error! Bookmark not defined.Error! Bookmark not defined.Error! Bookmark not defined..** As NFHS-2 only collected data on ever-married women, we then restricted the sample across surveys to this population, to allow comparability over time. Pregnant women were also not included in our analysis as their pregnancy may bias their assessment of weight status. **From this restricted sample, we further excluded women (1998-99: n=6182 (7.4%); 2005-06: n=3673 (4.2%); 2015-16: 7810 (1.6%)) and men (2005-06: n= 5160 (6.8%); 2015-16: n=3422 (3.1%)) with missing height and weight data. The analytic sample used in our main analysis consisted of 628,795 women aged 15–49 years and 93,618 men aged 15-54 across all three surveys, representing respondents with complete data across all the key variables.** In each of the surveys, multi-stage sampling approaches were adopted, and sampling weights were provided in the data sets. **Error! Bookmark not defined.Error! Bookmark not defined.Error! Bookmark not defined..**

References

Subramanian, S.V. and Smith, G.D., 2006. Patterns, distribution, and determinants of under-and overnutrition: a population-based study of women in India–. *The American journal of clinical nutrition*, 84(3), pp.633-640.

Subramanian, S.V., Perkins, J.M. and Khan, K.T., 2009. Do burdens of underweight and overweight coexist among lower socioeconomic groups in India?–. *The American journal of clinical nutrition*, 90(2), pp.369-376.

P6

Table 1 The rural/urban split hasn't been presented in isolation (only in conjunction with the SoL data). I think it would be helpful to see this data separately.

Response: Many thanks for this comment. Given the stratification by urban and rural, we agree this data presented separately would be helpful. These tables were prepared and included in the document as the new Tables 1 and 2.

P9 Discussion

L12 I think there is a typo '...overweight/obesity is generally higher among never-married women..'. Should this read lower?

Response: Yes, thank you for spotting this error. This typo has been changed.

"...overweight/obesity is generally **lower** among never-married women..."

Reviewer: 2

Reviewer Name: Dr. Tehzeeb Zulfiqar

Institution and Country: Australian National University, Australia

Please state any competing interests or state 'None declared': None declared

Please leave your comments for the authors below

It is a very important study as there are gaps in evidence from low-and-middle-income countries about obesity prevalence.

Response: Thank you for your positive feedback.

1. Introduction. Para 1 line 24 & 3 line 43. The term over nutrition is confusing. Obesity would be more appropriate term to use here.

Response: The term over-nutrition has been altered to overweight and obesity where specified.

2. Please use one term for overweight/obesity; either use overweight or obese OR overweight/obese OR overweight and obese. At present all three terms are used in the introduction.

Response: "...classified as overweight or obese..." has been chosen as the preferred term where appropriate. For example:

“In this study, we aim to estimate recent trends in **the proportion of Indians considered overweight or obese** by SEP in India.”

3. The term "waves" give an impression that NFHS is a longitudinal study. If so, then why longitudinal analysis was not done? If it is a repeated cross-sectional study then please use another term instead of waves.

Response: The term ‘wave’ has been changed to ‘survey’.

4. The second sentence of results (For instance, whereas the percentage of women in the sample classified as having no education...is quite confusing. please try to rewrite. In addition, does " the previous ones" in the end of the sentence mean the previous surveys?

Response: Thank you for this comment. In line with Helen Booth’s comments, Table 1 in the original document has been changed to the new Tables 1 and 2, presenting descriptive statistics for urban and rural populations separately. The sentence referred to in this comment refers to the newly inputted tables:

“Whereas the percentage of respondents with no education declined over the study period, particularly among the rural population, the percentage with secondary education in the 2015-16 survey was generally higher than in 1998-99 and 2005-06. Additionally, in both rural and urban areas, the percentage of individuals from lower SoL households declined, whilst the percentage from higher SoL households increased between 1998 and 2016 (Tables 1 and 2).”

5. Please revise the sentence in 2nd para line 33 of "potential mechanisms" for clarity (..some researchers have suggested that high prices of low calorie foods may price lower SEP individuals out of healthy diets)

Response: Thank you for the comment, this sentence has been reorganised to be clearer:

“that lower SEP individuals may be priced out of affording relatively expensive low-calorie healthy diets”

6. Reference 23 line 52 page 12, is in a different format than the rest. Please check.

Response: Thank you spotting this error. We have changed the reference to the same style as all the others:

Misra, A., Vikram, N.K., Gupta, R., Pandey, R.M., Wasir, J.S. and Gupta, V.P., 2006. Waist circumference cutoff points and action levels for Asian Indians for identification of abdominal obesity. *International journal of obesity*, 30(1), p.106.